# Dissolution of the Disparate: Co-ordinate Regulation in Antibiotic Biosynthesis

**DOI:** 10.3390/antibiotics8020083

**Published:** 2019-06-18

**Authors:** Thomas C. McLean, Barrie Wilkinson, Matthew I. Hutchings, Rebecca Devine

**Affiliations:** 1School of Biological Sciences, University of East Anglia, Norwich Research Park, Norwich NR4 7TJ, UK; t.mclean@uea.ac.uk (T.C.M.); r.devine@uea.ac.uk (R.D.); 2Department of Molecular Microbiology, John Innes Centre, Norwich Research Park, Norwich NR4 7UH, UK

**Keywords:** Secondary metabolism, regulation, biosynthesis, antibiotics

## Abstract

Discovering new antibiotics is vital to combat the growing threat of antimicrobial resistance. Most currently used antibiotics originate from the natural products of actinomycete bacteria, particularly *Streptomyces* species, that were discovered over 60 years ago. However, genome sequencing has revealed that most antibiotic-producing microorganisms encode many more natural products than previously thought. Biosynthesis of these natural products is tightly regulated by global and cluster situated regulators (CSRs), most of which respond to unknown environmental stimuli, and this likely explains why many biosynthetic gene clusters (BGCs) are not expressed under laboratory conditions. One approach towards novel natural product discovery is to awaken these cryptic BGCs by re-wiring the regulatory control mechanism(s). Most CSRs bind intergenic regions of DNA in their own BGC to control compound biosynthesis, but some CSRs can control the biosynthesis of multiple natural products by binding to several different BGCs. These cross-cluster regulators present an opportunity for natural product discovery, as the expression of multiple BGCs can be affected through the manipulation of a single regulator. This review describes examples of these different mechanisms, including specific examples of cross-cluster regulation, and assesses the impact that this knowledge may have on the discovery of novel natural products.

## 1. Introduction

Antimicrobial resistance is one of the greatest threats to modern healthcare, and the development of new drugs to treat drug-resistant infections is vital. The majority of antimicrobials currently used in the clinic are derived from the natural products of soil-dwelling actinomycetes, particularly *Streptomyces* bacteria [1]. These natural products are made as part of the normal secondary metabolism of these bacteria and are, therefore, also referred to as secondary metabolites. *Streptomyces* species have been studied as a source of natural product antimicrobials since the ‘golden age’ of antibiotic discovery, with the first treatment for tuberculosis (TB), streptomycin, described from *Streptomyces griseus* in 1943 [2]. The increased availability of genome sequencing data in recent years has revealed that the majority of biosynthetic gene clusters (BGCs) coding for natural products in these actinomycete bacteria are not expressed under standard laboratory conditions and, therefore, there is a vast amount of chemical diversity that remains to be explored [3]. These silent or cryptic BGCs are more likely to be a source of compounds with novel structures and novel mechanisms of action and are thought to hold promise as treatments against drug-resistant infections [4]. *Streptomyces* species and other antibiotic-producing actinomycetes have a complex and multicellular developmental life cycle that is tightly controlled by an intricate regulatory network, much of which remains largely uncharacterised. Moreover, it is well documented that the control of growth and development in *Streptomyces* species is strongly coupled to the regulation of secondary metabolite production [5,6]. By understanding the complex regulatory networks that control the expression of secondary metabolite BGCs, the potentially novel chemistry encoded could be accessed for clinical development [7]. 

Secondary metabolite production begins with the transcription of the biosynthetic genes [8]. During transcription, the RNA polymerase (RNAP) transcribes the DNA into messenger RNA (mRNA), which can then be translated into the functional protein by the ribosome [9]. To initiate transcription of the biosynthetic genes, the RNAP must, therefore, recognise the promoter sequences within a BGC. This process can be enhanced or repressed by transcriptional regulator proteins. On binding of a transcriptional activator or the release of a transcriptional repressor, the RNAP is able to recognise the promoter and begin the process of transcription. Most secondary metabolite BGCs contain all the genes required for the synthesis of the compounds in one or more neighbouring operons. In addition, genes coding for host-resistance and one or more regulators are often also contained within the BGC [5]. Regulator(s) encoded within the BGC are known as cluster situated regulators (CSRs) and usually control the expression of the biosynthetic genes within the same BGC. However, the expression of a BGC might also be under the control of one or more global regulatory systems that sense environmental signals and link secondary metabolism with the primary metabolism and other aspects of fundamental growth and development [3,7]. Furthermore, cross-cluster regulation, where a CSR from one cluster is involved in the regulation of another BGC elsewhere in the genome, can also occur and is likely to be more widespread than current reports suggest. 

### 1.1. Cluster Situated Regulators (CSRs) Control the Expression of Biosynthetic Genes within their Pathways

One of the largest families of CSRs is the *Streptomyces* antibiotic regulatory protein (SARP) family. SARP homologs are encoded by many BGCs, including type I and type II polyketide synthases (PKS), non-ribosomal peptide synthases (NRPS), ß-lactam and lantibiotic gene clusters [7]. These regulators are characterised by an N-terminal winged helix-turn-helix (HTH) motif responsible for DNA binding [5]. SARPs usually activate the expression of biosynthesis genes by binding to heptameric repeat sequences in the promoter regions of the genes they regulate [7]. Some of the most well characterised secondary metabolite BGCs are those found in the model actinomycete *Streptomyces coelicolor* [10]. The BGCs coding for the antibiotics actinorhodin, undecylprodigiosin and the calcium-dependent antibiotic (CDA) in *S. coelicolor* all contain SARP family regulators that activate the transcription of the biosynthetic genes within the cluster; ActII-orf4, RedD and CdaR, respectively [11]. In addition to the SARPs, the second major class of positive CSRs are the large ATP-binding regulators of the LuxR family (LAL) proteins. While SARPs are found specifically in actinomycete genomes, genes coding for LAL homologues have been found in other phyla, including the Gram-negative proteobacteria [7]. LAL regulators usually contain an N-terminal nucleotide triphosphate (NTP) binding motif in addition to a C-terminal HTH motif. A well-characterised example is the LAL regulator PikD, which is the primary activator of the macrolide biosynthesis genes in the multi-drug encoding polyketide synthase (PKS) BGC in *Streptomyces venezuelae,* which encodes for the biosynthesis of methymycin, neomethymycin, narbomycin and pikromycin [12]. Similarly, *rapH* encodes a LAL regulator in *Streptomyces hygroscopicus* that plays a role in the activation of the biosynthesis of rapamycin, a macrocyclic polyketide with a broad range of bioactivities [13]. 

Many CSRs have ligand binding capabilities in addition to binding DNA, and they alter gene expression levels within a BGC in response to the presence of cellular metabolites. For example, TetR-family regulators are a major class of repressor involved in adaptive responses that typically contain a 47 amino acid long HTH DNA binding motif and a ligand binding domain on the same polypeptide [3,14]. The binding of the ligand, often one or more products of the biosynthesis pathway, induces a conformational change that inhibits the interaction with DNA [7]. In this way, TetR regulators can alter the transcription of genes in the BGC in response to specific signals and are sometimes referred to as one component systems. Well-characterised examples of TetR regulators include the JadR and JadR2 regulators of jadomycin biosynthesis (discussed in more detail below) [11]. Cluster situated TetR-family regulators often regulate the expression of the self-resistance mechanism, for example, by controlling the expression of efflux pumps, as well as autoregulating their own expression [14]. Many other regulator protein families, such as the LacI- and MerR-family regulators, also function as one component systems, with one domain responsible for DNA binding and another either binding a ligand or sensing a stressor, to allow these CSRs to respond to stimuli, such as oxidative stress, heavy metals or antibiotics [15]. In contrast, MarR-family transcriptional regulators bind DNA and their ligands using the same domain. They are found throughout bacteria and are named after the multiple antibiotic resistance regulator in *Escherichia coli.* MarR proteins function as homodimers where each monomer contributes a winged HTH DNA binding motif, and they usually repress transcription of their target genes, often by binding to palindromic repeat sequences in the DNA. Like other ligand-responsive transcriptional repressors, the binding of a ligand blocks the protein-DNA interaction and induces gene expression. MarR regulators often control the expression of host-resistance mechanisms and efflux pumps, and the ligand is usually a product of the biosynthetic pathway they control [16]. An example is the MarR family transcriptional regulator SAV4189, encoded within the avermectin BGC in *Streptomyces avermitilis*, which represses its own transcription and an adjacent, co-transcribed gene coding for an efflux pump, which is presumably involved in removing avermectin from the cell. SAV4189 has also been shown to indirectly activate avermectin production through interactions with another CSR, AveR [17].

### 1.2. Antibiotic Biosynthesis is Controlled by Both High- and Low-Level Regulatory Systems

CSRs are traditionally considered to be at the bottom of the regulatory cascade and involved in the specific regulation of a single BGC, but some have been shown to cause pleiotropic changes in gene expression across the genome of antibiotic-producing microorganisms. For example, ActII-orf4, the SARP-family CSR of the actinorhodin BGC in *S. coelicolor,* has been shown to influence the transcription of genes in multiple other BGCs across the genome of *S. coelicolor*. Constitutive overexpression of *actII-orf4* causes changes to the levels of *cda* transcripts and decreases the expression of genes in the undecylprodigiosin (*red*) BGC [18]. Furthermore, the CSRs ActII-orf4 and RedZ, another CSR in the *red* BGC, have been shown to alter transcription levels of genes from the *whiE* spore pigment BGC and the NRPS BGC coding for coelichelin siderophore biosynthesis, as well as genes involved in antioxidant production in *S. coelicolor.* This illustrates that the regulation of secondary metabolism by these ‘low-level’ regulatory proteins is more complex than previously thought [18,19]. Although mostly coordinated by global regulators, some CSRs are able to respond to extracellular signals or metabolites that are produced by other biosynthetic pathways within the cell. For example, LysR-family transcriptional regulators consist of a highly conserved N-terminal HTH motif for DNA binding and a more variable C-terminal region for co-factor binding. They are highly conserved among bacteria and are usually CSRs, but some can also act as global transcriptional regulators, and they have been implicated in diverse functions, including the regulation of metabolism, cell division and oxidative stress [20]. An example is the ClaR regulator of the clavulanic acid biosynthesis pathway in *Streptomyces clavuligerus.* Deletion of *claR* results in changes to transcription of genes from the clavulanic acid BGC, as well as changes to the transcription of genes within the disparately located cephamycin C and holomycin BGCs. Deletion of *claR* also affects the transcription of higher-level regulator genes in the genome that are not related to secondary metabolite biosynthesis, such as those involved in the formation of aerial mycelium and control of oxidative stress, causing global changes to transcription [21].

Many global regulators have also been shown to play an important role in the control of secondary metabolite production, either directly or indirectly through the regulation of CSR activities. The Bld regulators control developmental processes, such as the generation of aerial hyphae, but some members of this family also regulate secondary metabolite biosynthesis [5]. BldD is well-known for its role in controlling entry into sporulation by forming a complex with cyclic di-GMP. BldD also indirectly affects antibiotic biosynthesis by interacting with BldC, which is a small DNA-binding protein belonging to the MerR family of transcriptional activators, that is involved in regulating the expression of genes from both the actinorhodin and undecylprodigiosin BGCs, as well as also controlling morphological differentiation, specifically the development of aerial mycelium [22]. Furthermore, BldD has also been shown to bind promoters within some BGCs like the erythromycin cluster in *Streptomyces erythraea* to directly control the expression of biosynthetic genes [23,24,25]. The GntR-family is another group of global regulators that consist of a typical N-terminal HTH DNA binding domain and a C-terminal effector-binding and oligomerisation domain [26]. Members of this family, such as DasR, have been shown to control the expression of genes involved in both morphogenesis and antibiotic production in response to environmental signals and the availability of nutrients [27]. DasR alters the levels of secondary metabolite gene expression and antibiotic production in response to the levels of N-acetylglucosamine in the environment [28]. In *S. coelicolor*, DasR indirectly controls the production of both actinorhodin and undecylprodigiosin by repressing the transcription of the CSRs ActII-orf4 and RedZ. Similarly, in *Saccharopolyspora erythraea*, the industrial producer of erythromycin A, deletion of *dasR* has been shown to decrease erythromycin production. However, the erythromycin A BGC does not encode a CSR; therefore, DasR must regulate erythromycin production via a different mechanism, possibly by directly binding promoter regions within the cluster [29]. AraC-family transcriptional regulators also have two domains; an N-terminal dimerisation and ligand binding domain and a highly conserved DNA binding C-terminal HTH domain. They are involved in the control of carbon source utilisation, morphological differentiation, secondary metabolism, pathogenesis and stress responses [30]. A good example is AdpA, a transcriptional regulator in *Streptomyces griseus,* that activates the transcription of genes involved in morphological development and secondary metabolism in response to the microbial hormone known as A-factor [31]. Another example is SAV742 in *S. avermitilis*, which controls avermectin biosynthesis, cell growth and morphological differentiation [30]. 

While one component systems contain ligand and DNA binding domains within a single protein, two-component systems (TCS) use separate sensor and regulator proteins to perceive and respond to environmental signals. A typical TCS consists of a transmembrane sensor histidine kinase and an associated cytoplasmic response regulator, with the kinase responsible for sensing the signal, undergoing autophosphorylation and then transferring that phosphate to its cognate response regulator. The majority of response regulators belong to the OmpR/PhoB subfamily and consist of a conserved α/β receiver domain and a C-terminal effector domain for DNA binding. Antibiotic-producing bacteria, such as *Streptomyces* species, generally encode for large numbers of TCS to allow them to control secondary metabolism and many other fundamental aspects of their life cycle in response to extracellular signals, such as nutrient deprivation and microbial competition [32]. Some BGCs encode cluster-situated TCSs, which, although not well characterised, are likely responsible for the control of expression of biosynthetic genes within the same BGC. An example is the CinKR TCS encoded in the cinnamycin BGC in *Streptomyces cinnamoneus* DSM 40646, which increases the expression of genes within the cinnamycin BGC to increase compound production once the host-resistance mechanism is induced to prevent the deleterious accumulation of the antibiotic [33]. Other TCS, like MtrAB, are involved in the global regulation of secondary metabolite production and coordinating this with growth and development [34,35]. *Streptomyces* TCSs are reviewed in detail elsewhere [36].

Thus, most BGCs encode one or more CSRs, but secondary metabolism is also under the control of higher regulatory systems, including one component systems like Crp and DasR and TCS like MtrAB. These examples begin to illustrate the complexity of regulatory systems that control bacterial secondary metabolism. Another level of complexity comes from CSRs like ActII-orf4, which appear to cross-regulate other BGCs elsewhere in the genome (Figure 1). These systems can control the levels of production of two or more secondary metabolites (Table 1).

## 2. Coordinate Regulation of BGCs

### 2.1. FscRI—Antimycins and Candicidin

The antimycins are depsipeptide natural products, which inhibit cytochrome *c* reductase and exhibit broad-spectrum bioactivity against a wide range of organisms (Figure 2). They are commercially used in the catfish industry as a pesticide to remove unwanted fish (catfish are resistant), and they also exhibit antifungal, insecticidal and nematocidal activity. Antimycins are being investigated as chemotherapeutic adjuvants due to their ability to inhibit the anti-apoptotic proteins Bxl-2/Bcl-2X_L_. These proteins are overproduced by certain cancers and confer resistance to apoptotic chemotherapy drugs [37]. Candicidin is a polyene antibiotic named after its efficacy against *Candida* that was once used to treat vaginal candidiasis and prostatic hyperplasia (Figure 2) [44]. Both candicidin and antimycins are produced by the leafcutter ant-derived strain *Streptomyces albidoflavus* S4, previously reported as *Streptomyces albus* S4 [45]. Candicidin biosynthesis is regulated by the CSRs FscRI and FscRIV, which, in turn, activate the expression of genes in the same cluster that encodes two more CSRs, FscRII and FscRIII. FscRI shares 69% identity to PimM, the PAS-LuxR CSR, which positively activates biosynthesis of the tetraene macrolide antifungal pimaricin in *Streptomyces natalensis* [46,47]. It was reported previously that heterologous overproduction of PimM in *Streptomyces albus* J1074 led to increased antimycin production, this is curious as it shows possible degeneracy in the activity of these PAS-LuxR family regulators [48]. Deletion of *fscRI* in *S. albidoflavus* S4 abolished both antimycin and candicidin production, and ChIP-seq analysis identified binding sites for FscRI upstream of genes in both the antimycin and candicidin BGCs: *antBA, antCDE*, *fscA, fscB1, fscB2, fscD* and *fscMI* [37]. The gene *antA* encodes the orphan ECF σ^AntA^, which is a transcriptional activator for the remaining antimycin biosynthetic genes (*antFG* and *antHIJKLMNO*) that are not activated by FscRI [49]. The *S. albidoflavus* S4 *ant* gene cluster was introduced into the *S. coelicolor* heterologous host M1146, but antimycins were only detected when *fscRI* was also expressed in this strain. There are three forms of *ant* gene cluster: short-form (S-form), intermediate-form (I-form) and long-form (L-form). *S. albidoflavus* S4 contains an S-form *ant* cluster, and the conservation of FscRI binding sites is limited to genomes containing the *ant* S-form cluster. Given their co-regulation, the synergism of antimycin and candicidin activities was tested using fractional inhibitory concentration (FIC) assays. No synergy of anti-fungal activity was detectable against *Candida albicans* CA6, but they exhibit synergy against *Escovopsis weberi*, the co-evolved fungal pathogen of the leafcutter ant’s nest [50]. Whilst antimycin resistance is conferred by a single point mutation in cytochrome *c* reductase, resistance to polyene antifungals, such as candicidin, comes at a significant fitness cost via the alteration of sterol biosynthesis [37].

### 2.2. CcaR—Cephamycin C and Clavulanic Acid

*Streptomyces clavuligerus* produces both cephamycin C, a ß-lactam antibiotic, and clavulanic acid, a ß-lactamase inhibitor [51,52]. The BGCs encoding these natural products are adjacent to each other in the genome forming a 35 kbp ‘super-cluster’, and although structurally related, their biosynthetic precursors are different [53]. Their expression is coordinately controlled by a SARP-type CSR called CcaR encoded in the cephamycin C BGC [54]. CcaR is similar to regulatory proteins, such as ActII-orf4 and RedD, which regulate the production of actinorhodin and undecylprodigiosin in *S. coelicolor*. Disruption of CcaR abolished both cephamycin C and clavulanic acid production in *S. clavuligerus,* while overexpression of *ccaR* led to a twofold increase in the production of cephamycin and a threefold increase in clavulanic acid after 48 hours [38]. Using purified CcaR protein, it was possible to identify heptameric repeats, which are putative binding sites for CcaR. These coincided with the promoter regions of multiple genes in both the cephamycin C and clavulanic acid BGCs and were confirmed using electrophoretic mobility shift assays (EMSAs). RT-PCR and qRT-PCR revealed large polycistronic mRNAs, including *ceaS2,* which encodes the carboxyethylarginine synthase CeaS2, responsible for committing arginine to clavulanic acid biosynthesis. The transcript levels of this mRNA were 1000–10,000× lower in the *ccaR* disruption mutant. A similar effect was seen for the cephamycin C pathway with the proteome of the mutant lacking multiple important biosynthetic proteins, including PcbC and CefF, the isopenicillin N synthase and deacetoxycephalosporin C hydroxylase, respectively. These changes, and others reported by Santamarta et al., account for the lack of production of both final products in the ∆*ccaR* mutant and reveal the coordinate regulation of cephamycin C and clavulanic acid in *S. clavuligerus* by CcaR [55]. 

On its own, clavulanic acid has little in vitro antimicrobial activity, but it has proven effective in combination with the semi-synthetic antibiotic amoxicillin for the treatment of acute otitis media, an infection of the middle ear. High-dose formulations are also used to treat penicillin-resistant *Streptococcus pneumoniae* [56,57]. The synergism arises from the ability of clavulanic acid to protect ß-lactam antibiotics, such as amoxicillin, from intra- and extracellular ß-lactamases [58]. It is reasonable to assume that the co-ordinate production of cephamycin C alongside clavulanic acid is to produce a similar effect as a result of the molecular arms race involving the evolution of protective ß-lactamases by *S. clavuligerus* competitors.

### 2.3. GdmRIII

The macrodiolide antibiotic elaiophylin (Figure 2) was originally identified from *Streptomyces melanosporus* for its in vitro antimycobacterial activity [59] and has since attracted interest as a potential antitumor drug due to its inhibition of autophagy [60]. It is often found co-produced with geldanamycin, a benzoquinone ansamycin antibiotic first found in *Streptomyces hygroscopicus,* which also displays antitumor activity, through binding to and inhibiting the chaperone activity of eukaryotic Hsp90 [39,61]. In *Streptomyces autolyticus,* the geldanamycin BGC encodes the LuxR-family regulators, GdmRI and GdmRII, and the TetR-family regulator, GdmRIII. In an *S. autolyticus* ∆*gdmRIII* mutant, the production of geldanamycin was significantly reduced. Three other compounds, elaiophylin and its congeners (11′-*O*-methylelaiophylin and 11,11′-*O*-dimethylelaiophylin), were markedly increased. Quantitative RT-PCR revealed that compared to the wild-type, the ∆*gdmRIII* mutant had reduced levels, some undetectable, of multiple geldanamycin biosynthetic gene transcripts, including *gdmRI, gdmRII* and the tailoring enzyme genes (*gdmF, M, N, H, K and gdmP)*. The disruption of *gdmRIII* had the opposite effect on the elaiophylin transcriptome with a marked increase in major biosynthetic genes, including *elaE, F, G and elaO,* but a slight decrease in the *elaI* exporter gene transcript was also observed. EMSAs using purified GdmRIII confirmed in vitro binding to the promoter regions of *gdmM, gdmN* and *elaF*. DNase I footprinting revealed the conserved non-palindromic binding sequence (5′-ATNGAGGAC-3′). TetR family regulators often bind to palindromic regions, but the binding of non-palindromic regions has also been reported [39].

Whilst these two antibiotics are co-produced, GdmRIII has an inverse effect on geldanamycin and elaiophylin, positively regulating the former and repressing the latter, although not completely. No studies into possible reasons for this co-production have been reported, although it has been postulated that the inverse regulation is important to control the flux of precursor metabolites. Both malonyl-CoA and methylmalonyl-CoA are utilised as substrates by both the geldanamycin and elaiophylin PKSs; thus, it is possible that GdmRIII works under nutrient-limited conditions to selectively synthesise the preferred compound [39].

### 2.4. JadR1

The model organism *Streptomyces venezuelae* has been shown to synthesise two antibiotics in vivo, chloramphenicol and the jadomycins (Figure 2). Chloramphenicol is a broad-spectrum antibacterial, active against both Gram-positive and Gram-negative bacteria and inhibits protein synthesis by binding to the 50S ribosome subunit [40,62]. The jadomycins are angucycline antibiotics with a cytotoxic activity whose expression is induced via ethanol, phage or yeast co-culture shock when grown in minimal media [63]. Regulation of jadomycin biosynthesis by CSRs is complex, but the current model begins with the gamma-butyrolactone SVB1 modulating the activity of the γ-butyrolactone receptor JadR3, which activates transcription of the gene encoding the OmpR-type atypical response regulator *jadR1* and represses *jadR2*. JadR*, a TetR-family regulator, and JadR2 act together to repress *jadR1* transcription [40,64]. Disruption of the pseudo-γ-butyrolactone receptor JadR2 led to the ethanol-stress independent expression of jadomycin, revealing JadR1 as a positive activator of the *jad* cluster and a repressor of chloramphenicol biosynthesis [40]. JadR1 is an atypical OmpR-family response regulator as it lacks the residue needed for phosphorylation [65]. The ∆*jadR1* mutant produced no detectable jadomycin, with or without ethanol stress, but chloramphenicol was significantly more abundant in the absence of ethanol when compared to the wild-type. Band shift assays revealed JadR1 binding to the *cmlI-cmlJ* intergenic region, which is vital for chloramphenicol production. Using S1 nuclease mapping, the levels of *cmlJ* mRNA were measured and found to be much higher in the *jadR1* mutant than the wild-type, confirming that JadR1 acts as a repressor of chloramphenicol biosynthesis. Interestingly both jadomycin and chloramphenicol can bind to JadR2 and prevent *jadR1* transcriptional repression, adding a further level of complexity to the regulation of these antibiotic gene clusters, but the overall effect is that JadR1 coordinately regulates jadomycin and chloramphenicol biosynthesis in an inverse manner [40].

The most obvious role of antibiotic production is to enhance the competitiveness of an organism in its given niche. However, there is growing evidence to suggest that they also act as important signalling molecules, and this example supports this idea. Whilst both jadomycin and chloramphenicol possess antibiotic activity against similar organisms, they also create an intra- and inter-BGC regulatory circuit and likely have effects as quorum-sensing compounds [64].

### 2.5. Phloroglucinol/PltM/PltR

Coordinate regulation of multiple secondary metabolite BGCs is not limited to streptomycetes. The coordinate production of 2,4-diacetylphloroglucinol (DAPG) and pyoluteorin by *Pseudomonas protegens* presents an interesting example of chemical coordinate regulation (Figure 2)**.** Both these metabolites have broad-spectrum antibiotic activity against bacteria, fungi, oomycetes and plants [41]. The chlorinated phenylpyrrole pyoluteorin is most notable for its activity against the plant pathogen *Pythium ultimum,* which is a broad range pathogen of plants, including crop species, such as corn, soybean and wheat [66,67,68]. DAPG is a polyketide metabolite produced by *Pseudomonas* spp. in the rhizosphere of wheat where it suppresses the fungal pathogen *Gaeumannomyces graminis* var. tritici, which causes wheat take-all disease [69,70,71]. 

Although many *Pseudomonas* strains have been shown to produce one or other of these antibiotics, *P. protogens* Pf-5 is among a small subset of strains, which is known to produce both DAPG and pyoluteorin. The biosynthetic genes are encoded in distinct clusters separated by 3.7 MB, and their production is tightly coordinated; deletion of the pyoluteorin biosynthetic genes results in overproduction of DAPG in addition to the loss of pyoluteorin [72,73]. Introduction of pyoluteorin to *P. protogens* bacterial cultures leads to the repression of DAPG production. The addition of the DAPG biosynthetic intermediate phloroglucinol (PG) at nanomolar concentrations activates pyoluteorin production; however, micromolar concentrations leads to repression [74,75]. The co-regulating factor, in this case, is not a protein but a (di)chlorinated metabolite intermediate. PG is processed by the pyoluteorin BGC-encoded halogenase PltM to form PG-Cl_2_ and PG-Cl, which directly bind to and activate the LysR-family regulator PltR, inducing transcription of the pyoluteorin BGC [41]. 

DAPG and pyoluteorin have been shown to function as antimicrobials and as cell-cell signalling molecules, whereas the intermediates, PG-Cl_2_ and PG-Cl, act as both intra- and inter-cellular signals orchestrating the activation or repression of DAPG and pyoluteorin production. DAPG is toxic to the bacterivorous amoeba *Acanthomoeba castellanii,* whilst pyoluteorin has no effect [76]. Antibiotic biosynthesis can be a heavy metabolic burden, reducing the general fitness of the producing organism. It becomes important to only produce the secondary metabolites when required, making repression as significant as activation [41,77]. 

### 2.6. Crp

The examples above focus on coordinate regulation between two BGCs, and there are also global regulators, which have a more pleiotropic effect. The Cyclic AMP receptor protein (Crp) has, barring *Firmicutes,* been found throughout the bacterial phyla [78]. In *Escherichia coli,* Crp and its effector molecule cAMP affect the expression of hundreds of genes mediating general carbon catabolite repression (CCR). Crp has over 70 different DNA binding sites in *E. coli,* and while it does not appear to have the same CCR effect in *Streptomyces* spp., it still plays an important role in global regulation as *S. coelicolor* ∆*crp* mutants display distinct morphological and developmental defects [79,80]. Crp homologues are also highly conserved (>90% amino acid sequence identity) among streptomycetes [42]. In addition to developmental homeostasis, Crp also coordinates regulation of multiple antibiotics, including ACT, RED, CDA and the yellow-pigmented polyketide antibiotic yCPK (Figure 2). These antibiotics, particularly the pigmented metabolites, are distinctive markers of secondary metabolism and one of the reasons why *S. coelicolor* has been the favoured model organism for five decades. ACT, RED, CDA and yCPK are regulated by the CSRs ActII-4, RedD/RedZ, CdaR and CpkO, respectively, which are, in turn, controlled individually or together by global regulators, such as MtrA, which will be discussed later [43]. ChIP-seq analysis of Crp combined with qPCR of the identified binding sites revealed that Crp binds directly to the promoters of all these CSRs. RT-qPCR of a ∆*crp* disruption mutant and the *crp* complement strain revealed further coordinate regulation with Crp inducing transcription of the NRPS gene cluster *SCO6429-38* and repressing the albaflavenone BGC. Overexpression of *crp* in the wild *Streptomyces* spp. strain WAC4988 resulted in the production of unidentified secondary metabolites with antibacterial activity, supporting the idea that Crp globally regulates secondary metabolism in streptomycetes [42]. 

Pleiotropic regulators, such as Crp, which seemingly modulate the production of numerous antibiotics, likely act as the canvas on which CSRs and other cellular regulatory elements act to add detail and fine-tuning to the secondary metabolome. The presence of certain carbon sources can require sweeping changes to both primary and secondary metabolism. Crp regulates the cellular response to such signals, which is particularly visible in the case of *S. coelicolor,* as Crp controls many of the conspicuous antibiotics, including ACT, RED, CDA and yCPK [42]. 

### 2.7. MtrA

The response regulator MtrA is highly conserved among Actinobacteria and forms one part of the MtrAB two-component system. It has been characterised as a master regulator in *S. coelicolor, S. venezuelae, Saccharopolyspora erythraea* and *Mycobacterium* species, including *M. tuberculosis,* from which its name originates (*M. tuberculosis* regulator A) and in which it is essential [81,82,83]. The activating signal of the sensor kinase MtrB remains unclear, but MtrA regulates numerous important genetic loci. In *M. tuberculosis,* MtrA has been shown to regulate expression of *dnaA* and *dnaN,* which encode for the DNA replicator activator and DNA clamp of DNA polymerase III, whilst also sequestering the chromosome origin *oriC* region. This sequestration of oriC is thought to prevent unwanted DNA replication during the post replication period and keep the cell cycle advancing towards division [84]. In *Streptomyces* spp., deletion of *mtrA* disrupts regular development, with mutants failing to produce aerial mycelium in a media-dependent manner [85]. 

Regarding antibiotic production, MtrA has been best studied in *S. coelicolor* and *S. venezuelae* where it is a pleiotropic regulator of secondary metabolism. In *S. coelicolor,* ChIP-seq of MtrA reveals binding upstream of *actII-orf4* and *redZ,* CSRs for the pigmented antibiotics ACT and RED. Whilst RedZ and ActII-orf4 are activators of their respective pathways, MtrA also binds between the divergent *actII-orf1* and *actII-orf2* genes, which encode a repressor and putative ACT transporter, respectively. This suggests that MtrA can directly modulate the antibiotic expression both positively and negatively. Deletion of the sensor kinase gene *mtrB* and, in effect, the loss of the regulatory function of MtrA result in the overproduction of both ACT and RED [43]. In *S. venezuelae,* a similar effect is observed with an ∆*mtrB* mutant producing >30× more chloramphenicol than the wild-type. ChIP-seq analysis showed MtrA binding between the divergent *cmlN* and *cmlF* genes, which encode for the chloramphenicol transporters required for production, and upstream of *clmR*, the CSR in this gene cluster [43,62]. MtrA was also shown to bind to the intergenic region between *jadR1* and *jadR2,* which, as discussed earlier, are involved in the regulation of jadomycin biosynthesis and antagonistic regulation of chloramphenicol. Thus, it is possible that MtrA could activate or repress jadomycin production in a similar manner to its regulation of ACT in *S. coelicolor* by selectively activating the transcription of either the repressor *jadR2* or the activator *jadR1* depending on the requirements of the cell at that time [35]. 

In streptomycetes, the production of secondary metabolites is intimately linked to the developmental transition from vegetative hyphae to aerial hyphae and the onset of sporulation [23,86]. Interestingly the *S. venezuelae ∆mtrB* mutant decoupled chloramphenicol production from this developmental schedule, resulting in constitutive production of the antibiotic. As MtrA has been shown to regulate cell cycle progression and development, it is likely that the MtrAB two-component system plays an important role in coordinating not only the production of multiple antibiotics but also the timing of antibiotic production with the correct developmental stages [35,43]. 

### 2.8. Studying Cross-Regulation of Secondary Metabolite BGCs Enables the Discovery of Novel Natural Products

Studying the regulation of secondary metabolite biosynthesis, especially cross-cluster regulation, has great potential value as we can use the knowledge gained to improve the production of secondary metabolites within strains of interest or even to discover new molecules from previously cryptic BGCs. Overexpressing pathway activators or inactivating pathway repressors is a well-established method of inducing the expression of cryptic BGCs and discovering new natural products [3,7,87]. However, with the recent increase in the availability of sequencing data, it can be difficult to know which cryptic BGCs to prioritise in the search for novel natural products, especially since it can be challenging to predict the products of clusters that show low homology to those in the database. It is well recognised that BGCs with low homology to other known clusters are likely to be a source of novel chemistry. If a single regulator can be identified that can control the production of more than one novel metabolite, it can increase the speed at which we can discover new molecules. For example, the filamentous fungi, *Aspergillus nidulans,* has been shown to encode 53 secondary metabolite BGCs, but the majority remain silent under normal laboratory conditions. One such cryptic cluster is the NRPS BGC, coding for the *inpA* and *inpB* genes, that is predicted to be responsible for the production of a tripeptide antibiotic. Overexpression of the putative activator at the *inp* locus, scpR, not only induces overexpression of the *inp* genes but also the upregulation of genes from the *afo* BGC and subsequent over-production of the novel polyketide metabolite asperfuranone. The biosynthesis of these two products is not linked, as deletion of the core NRPS genes from the *inp* cluster does not affect asperfuranon biosynthesis; therefore, it is predicted that scpR is able to control the expression of biosynthetic genes from both pathways due to the presence of a common motif within the intergenic regions of both gene clusters [88]. 

Studying regulation and coordinate regulation can also be used to inform on the biosynthesis of a molecule. For example, the need to cross-regulate multiple BGCs may indicate a biosynthetic link between two pathways, where one BGC synthesises the precursor molecule and enzymes from a distinct BGC elsewhere in the genome convert the molecule into the final product. For example, members of the fungal genus *Aspergillus* can combine polyketide biosynthesis with terpenoid precursors encoded by distinct BGCs to produce complex monoterpenoids that have a variety of biological activities [89]. The regulation of these clusters has not been studied in detail, but it is probable that there may be a single regulator that is common to both pathways. Examples such as CcaR, the regulator of both cephamycin C biosynthesis and clavulanic acid biosynthesis, can also inform how resistance mechanisms arise in antibiotic-producing bacteria and also how resistance may be transferred to pathogenic bacteria. They can also help to inform future treatment strategies; for example, the administration of β-lactamase inhibitors like clavulanic acid with amoxicillin is now widespread in the treatment of bacterial infections.

The limitation of using CSRs that coordinately regulate multiple BGCs as a tool for natural product discovery is that the genetic manipulation of the regulators usually needs to occur in the native host. Often, when non-model organisms are isolated that encode cryptic BGCs, the genes are captured and expressed in an organism that is easier to manipulate in the laboratory. However, on heterologous expression of a secondary metabolite BGC, any effects of cross-regulation by a CSR are lost as the other BGC(s) is/are likely not present. In the past, it has been challenging to genetically manipulate non-model actinomycetes and other antibiotic-producing microorganisms but recent advances, such as the development of widely applicable CRISPR/Cas9 protocols, has increased the efficiency with which these genetic mutations can be made [90,91]. Unfortunately, there remain multiple organisms, that are not genetically tractable even with CRISPR. While cryptic BGCs in these organisms can be studied through heterologous expression, the exploitation of cross-cluster regulation in these organisms will remain largely unexplored until improvements in genetic manipulation techniques can be made. 

In some instances, it is possible to exploit CSRs from within certain families of secondary metabolites in organisms other than the native host to increase the production of related natural products. For example, the glutarimide-containing polyketides iso-migrastatin, cyclohexamide and lactimidomycin are secondary metabolites with important biological activities, including cytotoxicity and antimicrobial activity, that are encoded by *Streptomyces platensis* NRRL 18993, *Streptomyces* sp. YIM56141and *Streptomyces amphibiosporus* ATCC 53964, respectively. The genes, *mgsA* and *chxA* are SARP-like regulators encoded within the iso-migrasratin and cycloheximide BGCs, respectively. No CSRs have been identified within the lactimidomycin BGC, but overexpression of either *mgsA* or *chxA* in *S. amphibiosporus* ATCC 53964 results in significantly increased titres of lactimidomycin [92]. Another example is the PAS-LuxR regulators of polyene macrolide biosynthesis, which are known to show high conservation that can be exploited to cross-regulate multiple antifungal BGCs. PimM was originally described as the pathway-specific regulator of pumaricin biosynthesis in *Streptomyces natalensis.* However, pimM has also been shown to increase production levels of candicidin, amphotericin and filipin when transformed into *Streptomyces albus, Streptomyces nodosus* and *Streptomyces avermitilis,* respectively, indicating that some regulators can be fully exchangeable between related BGCs [93]. Furthermore, heterologously expressed BGCs can be influenced by the regulators encoded within the host, and this can affect transcription of related secondary metabolite biosynthesis genes. For example, the ActII-orf4 regulator of the type II PKS actinorhodin BGC in *S. coelicolor* has also been shown to regulate the expression of a previously cryptic type II PKS BGC from *Streptomyces antibioticus* when the cluster is cloned into the former [94]. These examples show that the use of regulators that can affect the levels of gene transcription across multiple, disparate BGCs could prove a valid tool for genome mining and the discovery of novel natural products.

## 3. Conclusions

Coordinate regulation has made it increasingly clear that BGCs do not always form fully self-contained secondary metabolite production clusters as was once thought. The production of these compounds can be costly to the host, and they are often produced as effectors in response to specific signals. As such their production is tightly controlled by a network of multi-level regulators, from global regulators, such as Crp, to CSRs like FscRI, and can comprise the coordinate regulation of multiple BGCs within a single organism. It might be considered that a CSR that regulates more than one BGC is, in fact, a global regulator. However, we take the term global regulator to mean a transcription factor that regulates different types of genes and not just antibiotic biosynthesis genes within BGCs. Given the examples discussed above, the coordinate regulation of BGCs is likely to be a much more common occurrence than was previously considered, and its value should not be underestimated.

With the impending threat of an antimicrobial resistance crisis and the urgent need for new antimicrobials, significant effort has been made into the discovery of new antibiotic-producing BGCs. It may be possible to rewire entire BGCs to introduce constitutive promoters and express these metabolites; however, this may still prove difficult without understanding the regulatory elements found in other BGCs. Juxtaposed to this is the potential for vastly increasing our antimicrobial repertoire through the exploitation of coordinate regulation. Decoupling antagonistic regulatory elements (e.g., JadR1) would allow the expression of increased numbers of secondary metabolites, whilst heterologous expression of global regulators might lead to the expression of previously cryptic BGCs, such as those already been shown using Crp [40,42]. This review aims to provide a base of knowledge regarding the types and examples of antibiotic coordinate regulation. It also highlights the fact that the regulation of BGCs is complex and intertwined with the global regulation of primary metabolism via environmental signals, growth and the developmental program. This is further complicated by the binding of biosynthetic pathway intermediates and other chemical signals that may prove difficult to identify.

## Figures and Tables

**Figure 1 antibiotics-08-00083-f001:**
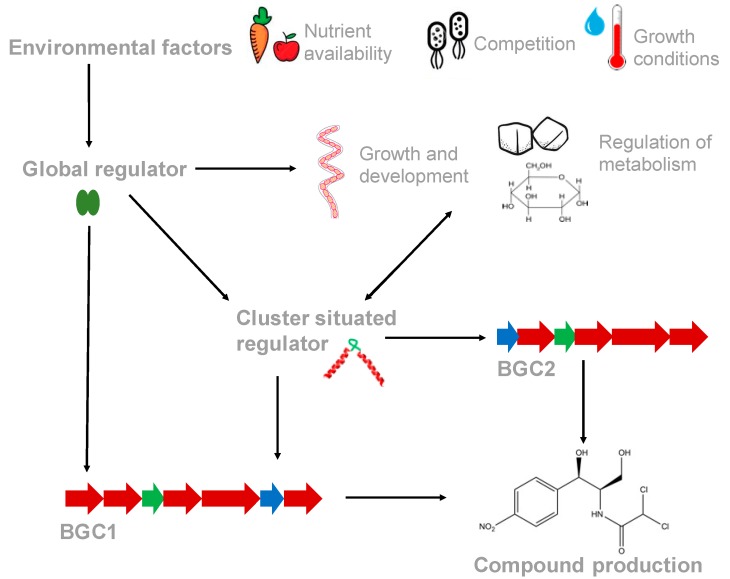
Regulation of secondary metabolite gene expression is controlled by cluster situated regulators (CSRs), which can be controlled by binding of pathway products, and global regulatory systems, which are influenced by external environmental factors, such as nutrient availability. Global regulators coordinate primary and secondary metabolism with growth and development. Global regulators can control the expression levels of CSRs, indirectly affecting BGC expression, or bind to elements within BGCs and directly control expression. Some CSRs are also influenced by (and/or can influence) primary metabolism, growth and development. Additionally, some CSRs can cross-regulate the expression of other BGCs in the genome, thereby controlling the levels of multiple secondary metabolites.

**Figure 2 antibiotics-08-00083-f002:**
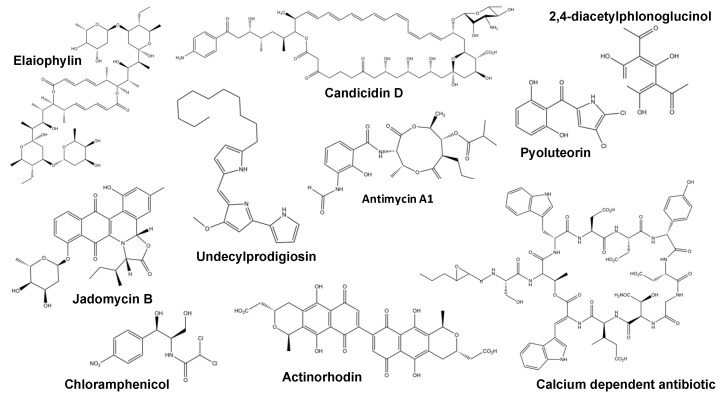
Structures of some of the compounds referenced in this review. The biosynthesis of these compounds is controlled by cross-cluster regulators.

**Table 1 antibiotics-08-00083-t001:** Comparison of the various coordinate regulatory systems described in this review.

Regulator Name.	Organism	Regulator Type	Antibiotics Regulated	Type of Regulation	Ref.
FscRI	*Streptomyces albus*	PAS-LuxR family	Candicidin	Positive direct	[37]
			Antimycins		
CcaR	*Streptomyces clavuligerus*	SARP-type	Cephamycin C	Positive direct	[38]
			Clavulanic acid		
GdmRIII	*Streptomyces autolyticus*	TetR-family	Geldanamycin	Antagonistic direct	[39]
			Elaiophylin		
JadR1	*Streptomyces venezuelae*	Atypical OmpR-family	Jadomycin	Antagonistic direct	[40]
			Chloramphenicol		
PG-Cl_2_/PG-Cl	*Pseudomonas protegens*	Chemical	2,4-diacetylphloroglucinol	Dose dependant	[41]
			Pyoluteorin		
Crp	*Streptomyces coelicolor*	Crp-Fnr family	Actinorhodin	Global	[42]
			Undecylprodiogisin		
			Calcium-dependant antibiotic		
			yellow-pigmented polyketide		
MtrA	*Streptomyces coelicolor*	OmpR-family	Actinorhodin	Global	[35,43]
			Undecylprodiogisin		
	*Streptomyces venezuelae*		Chloramphenicol		
			Jadomycin		

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
