# Peer review of "Dissolution of the Disparate: Co-ordinate Regulation in Antibiotic Biosynthesis"

_antibiotics, 2019, doi:10.3390/antibiotics8020083_

Round 1
Reviewer 1 Report
This is a nice review on the regulation of secondary metabolites biosynthesis focusing on examples of cross-cluster regulation. The review is concise, is well written, and the examples described are appropiately referenced. I thus recommend the acceptance of the draft in its current form.
Minor comments: Many CSRs seem to regulate more than one BGC, thus making the division between these CSR and global regulators somewhat blurred. Could these CSRs that exert control in the biosynthesis of more than one BGC be considered global regulators that happen to be encoded within the boundaries of one of the biosynthetic BGC? Or do the authors consider that there are significant differences between global regulatos and cross-cluster CSRs? Would a brief discussion on this subject be relevant in the final part of the manuscript?
Author Response
>We think there are significant differences, global regulators control many different genes not just BGCs but we have added a line to clarify this.
Reviewer 2 Report
Antibiotic resistance is a major challenge in healthcare these days so understanding their mechanism and studying the biosynthesis pathways to discover new natural antibiotic has become necessary to combat emerging resistance.
I find this review very interesting and relevant to understand the mechanism of antibiotic bio-synthetic pathways in antibiotic producing microorganism, their metabolites and regulatory systems. Its written well and it will definitely interest to the readers. I recommend this review to be published in antibiotics.
As a suggestion, I would like to add that if authors could represent some of the pathways and mechanism by schematic representation, it will make it more attractive and much easier for readers to understand, especially for those who are involved in interdisciplinary research.
Author Response
>We did attempt this but it was actually more confusing than leaving it out so we decided to add the schematic (Fig 1) instead.
Reviewer 3 Report
The manuscript is well organized and written.
The
manuscript by McLean and colleagues, entitled "Dissolution of the
disparate: co-ordinate regulation in antibiotic biosynthesis" is
focused on the review of regulators found associated with the control of
expression of genes involved in antibiotics biosynthesis. Global
regulators as well as regulators associated with the control of 2
antibiotic gene clusters are thoroughly reviewed throughout the
manuscript. Emphasis is given by authors to regulators that are involved
in the the regulation of multiple antibiotic operons. The authors claim
the importance of this knowledge in order manipulate the expression of
silent operons, and how this will lead in the future to the discovery of
novel antibiotics. The work is well organized and written. I have only a few minor comments to the manuscript. line 79: explain the abbreviations PKS and NRPS line 95: "encodes for proteins involved in the biosynthsis..." line 265: "anti-fungal activity was detected..." line 335: synthesize line 406: "binding sites in E. coli....". do the authors mean DNA binding sites? Please clarify.
Author Response
line 79: explain the abbreviations PKS and NRPS
>Changed
line 95: "encodes for proteins involved in the biosynthsis..."
>Changed
line 265: "anti-fungal activity was detected..."
>Changed
line 335: synthesize
>Changed
line 406: "binding sites in E. coli....". do the authors mean DNA binding sites? Please clarify.
>Yes, we have made this clear
